# DOMAIN GUIDANCE: A SIMPLE TRANSFER APPROACH FOR A PRE-TRAINED DIFFUSION MODEL

**Jincheng Zhong, Xiangcheng Zhang, Jianmin Wang, Mingsheng Long**[✉]
School of Software, BNRist, Tsinghua University, China
`{zjc22,xc-zhang21}@mails.tsinghua.edu.cn,`
`{jimwan,mingsheng}@tsinghua.edu.cn`

## ABSTRACT

Recent advancements in diffusion models have revolutionized generative modeling. However, the impressive and vivid outputs they produce often come at the cost of significant model scaling and increased computational demands. Consequently, building personalized diffusion models based on off-the-shelf models has emerged as an appealing alternative. In this paper, we introduce a novel perspective on conditional generation for transferring a pre-trained model. From this viewpoint, we propose *Domain Guidance*, a straightforward transfer approach that leverages pre-trained knowledge to guide the sampling process toward the target domain. Domain Guidance shares a formulation similar to advanced classifier-free guidance, facilitating better domain alignment and higher-quality generations. We provide both empirical and theoretical analyses of the mechanisms behind Domain Guidance. Our experimental results demonstrate its substantial effectiveness across various transfer benchmarks, achieving over a 19.6% improvement in FID and a 23.4% improvement in $FD_{DINOv2}$ compared to standard fine-tuning. Notably, existing fine-tuned models can seamlessly integrate Domain Guidance to leverage these benefits, without additional training. Code is available at this repository: `https://github.com/thuml/DomainGuidance`.

## 1 INTRODUCTION

Diffusion models have significantly advanced the state of the art across various generative tasks, such as image synthesis (Ho et al., 2020), video generation (Ho et al., 2022), and cross-modal generation (Saharia et al., 2022; Rombach et al., 2022). Concurrently, advancements in guidance techniques (Dhariwal & Nichol, 2021; Ho & Salimans, 2022) have significantly enhanced mode consistency and generation quality, becoming indispensable components of contemporary diffusion models (Esser et al., 2024; Peebles & Xie, 2023). However, generating high-quality samples frequently requires substantial computational resources to scale foundational diffusion models. In practical settings, transfer learning, especially fine-tuning, proves vital for personalized generative scenarios.

Recent research has yielded promising outcomes in fine-tuning scaled pre-trained models. Despite diverse motivations, these efforts converge on a common objective: efficient fine-tuning with minimal parameter adjustment, a group of methods termed parameter-efficient transfer learning (PEFT) (Houlsby et al., 2019; Zaken et al., 2021; Xie et al., 2023). Nevertheless, PEFT introduces significant optimization challenges, including the necessity for considerably higher learning rates—often an order of magnitude greater than typical—which may precipitate spikes in loss. An effective transfer strategy that capitalizes on the intrinsic properties of diffusion models remains largely unexplored.

In this paper, we introduce a novel perspective on conditional generation for fine-tuning. We conceptualize the transfer of a pre-trained model to a downstream domain as conditioning the sampling process on the target domain, relative to the pre-trained data distribution. From this viewpoint, we incorporate guidance principles (Dhariwal & Nichol, 2021; Ho & Salimans, 2022) and introduce *Domain Guidance* (DoG) as a general transfer method to enhance model transfer. DoG involves fine-tuning the pre-trained model specifically for the new domain to create a domain conditional branch, while simultaneously maintaining the original model as an unconditional guiding counterpart. At each sampling step, the domain conditional and the pre-trained guiding model are executed once

each, with the fine-tuned results being further extrapolated from the pre-trained base using a DoG factor hyperparameter. This method not only offers a general guidance strategy for transferring pre-trained models but also seamlessly integrates models fine-tuned in the classifier-free guidance (CFG) style by simply excluding the unconditional component, without necessitating additional training. This streamlined approach significantly improves domain alignment and generation quality.

To further explore the mechanism behind DoG, we provide both empirical and theoretical analyses. Firstly, we employ a mixture of Gaussian synthetic examples and perform a theoretical analysis of DoG behaviors, which reveal that DoG effectively leverages the pre-trained domain knowledge, improving domain alignment. In contrast, standard CFG with a fine-tuned model often suffers from catastrophic forgetting, eroding valuable pre-trained knowledge. Furthermore, we observe that limited training resources and a low-data regime typically challenge the unconditional guiding component's ability to fit the target domain, leading to out-of-distribution (OOD) samples and exacerbating sampling errors. DoG effectively mitigates these issues, reducing overall errors and enhancing the generation quality.

Experimentally, we evaluate DoG across seven well-established transfer learning benchmarks, providing quantitative and qualitative evidence to substantiate its efficacy. Our comprehensive ablation study further underscores its superiority in the transfer of pre-trained diffusion models.

Overall, our contributions can be summarized as follows:

- We introduce a novel conditional generation perspective for transferring pre-trained models and present *Domain Guidance* (DoG) as a streamlined, effective transfer learning approach that leverages the principles of CFG to enhance domain alignment and generation quality.
- We delve into the mechanisms behind DoG's improvements, offering both empirical and theoretical evidence that underscores how DoG enhances domain alignment by harnessing pre-trained knowledge. We also highlight how standard CFG approaches with fine-tuned guiding models often face challenges from poor fitness, which can exacerbate guidance performance issues due to increased variance in OOD samples. Conversely, DoG effectively addresses these concerns.
- We validate DoG across various benchmarks, confirming its effectiveness. Our quantitative assessments show marked improvements in generated image distributions, as measured by FID (Heusel et al., 2017) and $FD_{DINOv2}$ (Stein et al., 2024b) metrics, and reveal that existing fine-tuned models can benefit from DoG without additional training.

## 2 RELATED WORK

**Diffusion models.** Diffusion-based generative models (Ho et al., 2020; Song & Ermon, 2019; Song et al., 2020b; Karras et al., 2022) transform pure noise into high-quality samples through an iterative denoising process. This gradual transformation stabilizes the training process but also imposes substantial computational demands for sampling. Recent improvements in diffusion models have primarily addressed noise schedules (Nichol & Dhariwal, 2021; Karras et al., 2022), training objectives (Salimans & Ho, 2021; Karras et al., 2022), efficient sampling techniques (Song et al., 2020a), controllable generation (Ho & Salimans, 2022; Zhang et al., 2023; Dhariwal & Nichol, 2021), and model architectures (Peebles & Xie, 2023). Current state-of-the-art models benefit significantly from scaling up training parameters and datasets, necessitating considerable resources. In this work, we examine efficient transfer learning strategies for pre-trained diffusion models.

**Guidance techniques for diffusion models.** The notable successes of recent applications (Dhariwal & Nichol, 2021; Blattmann et al., 2023; Esser et al., 2024) using diffusion models can largely be attributed to advances in guidance techniques, which ensure that model outputs align closely with human preferences. Prior studies have developed various methods for effectively modeling conditional control information. Dhariwal & Nichol (2021) introduced *classifier guidance*, which enhances conditional generation through an additional trained classifier. Subsequently, *classifier-free guidance* (CFG), proposed by Ho & Salimans (2022), has emerged as the *de facto* standard in modern diffusion models due to its robust performance. Recently, a line of works Kynkäänniemi et al. (2024); Karras et al. (2024a) investigates how to utilize guidance techniques to enhance the generation performance in Elucidating Diffusion Models (EDM) Karras et al. (2022). Our work identifies challenges in the

underperformance of fine-tuned diffusion models within standard CFG frameworks and investigates novel guidance strategies for adaptation.

**Transfer learning.** Transfer learning seeks to leverage existing knowledge to facilitate learning in a new domain (Pan & Yang, 2009), typically through fine-tuning a pre-trained model (Yosinski et al., 2014). Previous research has aimed to refine standard fine-tuning techniques to address issues such as catastrophic forgetting (Zhong et al., 2024; Li & Hoiem, 2017), negative transfer (Chen et al., 2019), and overfitting (Dubey et al., 2018). With the recent significant expansion in model scales, the focus has shifted to a research area known as parameter-efficient transfer learning (Houlsby et al., 2019; Zaken et al., 2021), which aims to adjust as few parameters as possible to minimize memory usage and computational demands on gradient calculations. In this work, we reframe transfer learning in the context of domain conditional generation and propose a streamlined and effective approach.

## 3 METHOD

### 3.1 BACKGROUND

**Diffusion formulation.** Before demonstrating our method, we briefly revisit the basic concepts in diffusion models. Gaussian diffusion models are defined by a forward process that gradually adds noise to original samples: $\mathbf{x}_t = \sqrt{\alpha_t}\mathbf{x}_0 + \sqrt{1 - \alpha_t}\boldsymbol{\epsilon}$, where $\mathbf{x}_0 \sim \mathcal{X}$ denotes the original samples, $\boldsymbol{\epsilon} \sim \mathcal{N}(\mathbf{0}, \mathbf{I})$ denotes the noise signal, and constants $\alpha_t$ are hyperparameters that determine the level of noise infusion.

The training of diffusion models typically involves learning a parameterized function $f$ that predicts the noise added to a sample, formalized by the loss function:

$$L(\boldsymbol{\theta}) = \mathbb{E}_{t,\mathbf{x}_0,\boldsymbol{\epsilon}}\left[w_t\left\|\boldsymbol{\epsilon} - f_{\boldsymbol{\theta}}\left(\sqrt{\alpha_t}\mathbf{x}_0 + \sqrt{1 - \alpha_t}\boldsymbol{\epsilon}, t\right)\right\|^2\right], \tag{1}$$

where $w_t = 1$ is set by default, following the simple setting used in prior studies (Ho et al., 2020). Sampling from diffusion models $f_{\boldsymbol{\theta}}$ then follows a Markov chain, iteratively denoising from $\mathbf{x}_T \sim \mathcal{N}(\mathbf{0}, \mathbf{I})$ back to $\mathbf{x}_0$.

**Classifier-free guidance.** In various complex real-world scenarios, aligning the outputs of diffusion models with human preferences is crucial. Classifier-free guidance (CFG) has become an essential tool for enhancing the outputs of practically all image-generating diffusion models (Ho & Salimans, 2022; Esser et al., 2024; Karras et al., 2024b). CFG is formalized as follows:

$$\nabla_{\mathbf{x}_t} \log p_w^{\text{CFG}}(\mathbf{x}_t|c) = \nabla_{\mathbf{x}_t} \log p(\mathbf{x}_t|c) + (w - 1)\left(\nabla_{\mathbf{x}_t} \log p(\mathbf{x}_t|c) - \nabla_{\mathbf{x}_t} \log p(\mathbf{x}_t)\right). \tag{2}$$

Here, $w$ is the guidance factor, typically set greater than 1, modulating the influence between the outputs of the conditional and unconditional models to achieve the desired guiding effect.

Practically, CFG is implemented by constructing both a conditional model $\nabla_{\mathbf{x}_t} \log p_{\boldsymbol{\theta}}(\mathbf{x}_t|c)$ and an unconditional guiding model $\nabla_{\mathbf{x}_t} \log p_{\boldsymbol{\theta}}(\mathbf{x}_t)$ within a shared-weight network $f_{\boldsymbol{\theta}}$. The combined training loss is described as:

$$L(\boldsymbol{\theta}) = \mathbb{E}_{t,\mathbf{x}_0,\boldsymbol{\epsilon},c}\left[\left\|\boldsymbol{\epsilon} - f_{\boldsymbol{\theta}}\left(\sqrt{\alpha_t}\mathbf{x}_0 + \sqrt{1 - \alpha_t}\boldsymbol{\epsilon}, t, \text{Dropout}_\delta(c)\right)\right\|^2\right], \tag{3}$$

where the dropout ratio $\delta$ is typically set at 10%, as endorsed by recent studies (Peebles & Xie, 2023; Esser et al., 2024).

### 3.2 DOMAIN GUIDANCE

Fine-tuning existing checkpoints for target domains has become a prevalent practice in transfer learning. In this section, we introduce a novel perspective on transferring a generative model through the lens of conditional generation, bridging the commonly used classifier-free guidance (CFG) into transfer learning to develop our method, named *Domain Guidance* (DoG).

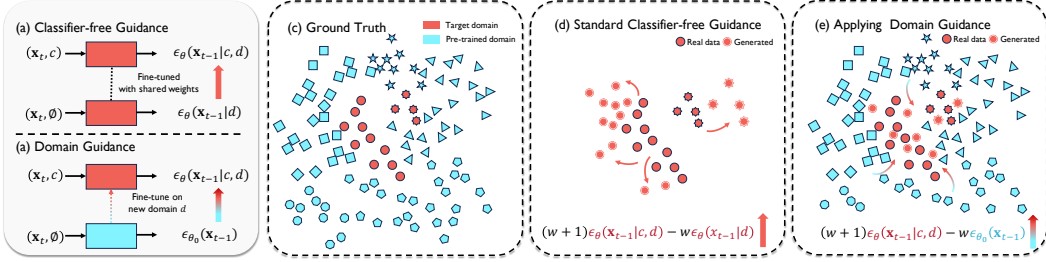

Figure 1: Conceptual comparisons between *Domain Guidance* and standard classifier-free guidance. (a) shows standard CFG modeling both conditional density and unconditional guiding signals for the target domain simultaneously. (b) illustrates the proposed *Domain Guidance*, which focuses on building conditional density and guides the sampling process from the pre-trained model to the target domain. (c) to (e) depict conceptual examples of the mechanism differences between CFG and DoG, highlighting how DoG leverages pre-trained knowledge to enhance generation for the target domain.

**The domain conditional generation perspective of transfer.** The primary goal in training a generative model on a target domain is to accurately capture its distribution. When fine-tuning a pre-trained generative model, we start with models that have learned the distribution of the pre-trained data. Ideally, it should leverage the distribution knowledge from the pre-trained data, effectively modeling the conditional distribution given this pre-trained context. However, the relationship $p(\mathbf{x}^{\mathrm{tgt}}|\mathcal{D}^{\mathrm{src}})$ is often compromised due to catastrophic forgetting, as the model loses access to the pre-trained data. Without adequate regularization from these pre-trained datasets, the model tends to converge solely to the marginal target distribution $p(\mathbf{x}^{\mathrm{tgt}}|\mathcal{D}^{\mathrm{tgt}})$ through empirical risk minimization with Equation 3. This convergence results in the loss of valuable pre-trained domain knowledge, limiting the standard fine-tuning effectiveness in modeling the relationship of the domain conditional generation.

**Guiding generations to the target domain via domain guidance.** Building on the domain conditional generation viewpoint, we introduce *Domain Guidance* (DoG), which utilizes the original pre-trained model as an unconditional guiding model. This approach leverages pre-trained knowledge to direct the generative process towards the target domain, as outlined below:

$$\boldsymbol{\epsilon}^{\mathrm{DoG}}(\mathbf{x}|\mathcal{D}^{\mathrm{tgt}}) = \boldsymbol{\epsilon}_{\boldsymbol{\theta}}(\mathbf{x}|\mathcal{D}^{\mathrm{tgt}}) + (w^{\mathrm{DoG}} - 1)\left(\boldsymbol{\epsilon}_{\boldsymbol{\theta}}(\mathbf{x}|\mathcal{D}^{\mathrm{tgt}}) - \boldsymbol{\epsilon}_{\boldsymbol{\theta}_0}(\mathbf{x})\right), \qquad (4)$$

where $\boldsymbol{\epsilon}_{\boldsymbol{\theta}}(\mathbf{x}|\mathcal{D}^{\mathrm{tgt}})$ represents the output of the fine-tuned model specific to the target domain, and $\boldsymbol{\epsilon}_{\boldsymbol{\theta}_0}(\mathbf{x})$ denotes the output from the original pre-trained model, with $\boldsymbol{\theta}_0$ marking the weights prior to fine-tuning. The guidance factor, $w^{\mathrm{DoG}}$, adjusts the influence of this guidance, where values greater than 1 typically emphasize traits of the target domain. Specifically, DoG reduces to the standard fine-tuned model output, $\boldsymbol{\epsilon}_{\boldsymbol{\theta}}(\mathbf{x}|\mathcal{D}^{\mathrm{tgt}})$, when $w = 1$ and to the pre-trained model $\boldsymbol{\epsilon}_{\boldsymbol{\theta}_0}(\mathbf{x})$, when $w = 0$.

DoG serves as a versatile mechanism for the transfer of a pre-trained model and can be directly expanded to a variety of transfer scenarios involving both conditional signals $c$ and domains $D^{\mathrm{tgt}}$, enhancing its applicability. The formulation of DoG in these contexts is given by:

$$\boldsymbol{\epsilon}^{\mathrm{DoG}}(\mathbf{x}|c, \mathcal{D}^{\mathrm{tgt}}) = \boldsymbol{\epsilon}_{\boldsymbol{\theta}}(\mathbf{x}|c, \mathcal{D}^{\mathrm{tgt}}) + (w^{\mathrm{DoG}} - 1)\left(\boldsymbol{\epsilon}_{\boldsymbol{\theta}}(\mathbf{x}|c, \mathcal{D}^{\mathrm{tgt}}) - \boldsymbol{\epsilon}_{\boldsymbol{\theta}_0}(\mathbf{x})\right), \qquad (5)$$

For practical implementation, inputs are concatenated with the conditional signal $c$, while the domain signal $D^{\mathrm{tgt}}$ is implicitly integrated during fine-tuning on the target domain. The dropout ratio $\delta$ in the standard CFG setup (Equation 2) can be set to 0, eliminating the need to fine-tune the unconditional guiding model and thereby simplifying the fitting process. Moreover, models that have been previously fine-tuned using the CFG approach can seamlessly transition to benefit from DoG by merely substituting the unconditional guiding component. This adjustment allows existing models to leverage pre-trained knowledge more effectively, enhancing their adaptability and performance in new domain settings.

**Comparision with CFG.** We conceptually compare DoG with CFG in Figure 1, illustrated within a general transfer scenario involving conditional signals $c$ and domains $d$. Both approaches exhibit distinct behaviors in transfer settings. The jointly fine-tuning with Equation 3 and performing CFG

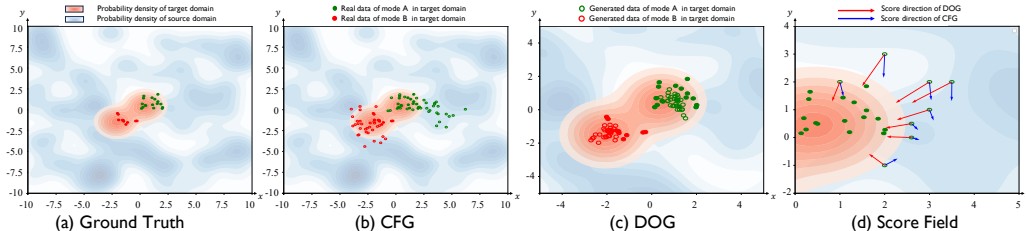

Figure 2: A mixture of Gaussians synthetic dataset with different colored dots represent modes of different classes. In (a), the target domain is defined by the orange area, while the pre-training distribution forms the blue background. Green and red dots represent two classes, with filled dots indicating in-domain real data.Sampling results from these classes after model fine-tuning are denoted by circles with corresponding color. (b) illustrates how CFG leads to out-of-domain samples by disregarding pre-trained knowledge, while (c) demonstrates how DoG maintains domain consistency by effectively utilizing pre-trained data. (d) contrasts the directional guidance provided by DoG (red arrows) against CFG (blue arrows) for intermediate samples $\mathbf{x}_{\text{mid}}$, showing how DoG steers samples towards the domain-specific regions, unlike CFG which may lead samples towards outliers.

on the fine-tuned model is the standard practice (e.g., (Esser et al., 2024; Xie et al., 2023; Zhang et al., 2023), as shown in Figure 1(a) and (d)). This method uses the target data to construct a weight-sharing network that models both conditional and unconditional densities simultaneously. Applying CFG through two forward passes can steer generation towards low-temperature conditional areas, thus enhancing generation quality and improving conditional consistency. However, CFG fails to leverage pre-trained knowledge due to catastrophic forgetting resulting from the inaccessibility of pre-trained data. As a result, directly performing CFG guides the generation to rely only on the limited support of the target domain, leading to high variance in density fitness and aggravating the OOD samples (as shown in Figure 1(d)). In contrast, DoG addresses these limitations by integrating the original pre-trained model as the unconditional guiding model, as depicted in Figure 1(b) and (e). DoG leverages the entire pre-trained distribution, which is typically more extensive than the target domain's distribution, to guide the generative process.

Remarkably, DoG can be implemented by executing both the fine-tuned model and the pre-trained model once each, thus not introducing additional computational costs compared to CFG during the sampling process. Unlike CFG, DoG separates the unconditional reference from the fine-tuned networks, allowing for more focused optimization on fitting the conditional density and reducing conflicts associated with competing objectives from the unconditional model. This strategic decoupling enhances the model's ability to harness pre-trained knowledge without the interference of unconditional training dynamics, leading to improved stability and effectiveness in generating high-quality, domain-consistent samples.

### 3.3 EMPIRICAL AND THEORETICAL INSIGHTS BEHIND DOMAIN GUIDANCE

We provide both empirical and theoretical evidence to demonstrate why Domain Guidance (DoG) significantly outperforms CFG when paired with standard fine-tuning. The advantages of DoG are primarily twofold: 1) DoG leverages pre-trained knowledge to guide generation within the target domain, achieving enhanced domain alignment, and 2) the unconditional guiding model in CFG often suffers from high variance due to underfitting in conditions of insufficient training and low-data availability in the target domain. This leads to an increased frequency of out-of-distribution samples.

**DoG leverages the pre-trained knowledge.** Building upon the conceptual differences illustrated in Figure 1, we analyze a 2D Mixture Gaussian synthetic dataset as a concrete example (Figure 2). This dataset features hundreds of modes, where a subset is designated as the target domain and the remainder as the source domain (shown in Figure This example consists of a mixture of Gaussian distributions with hundreds of modes, where a subset of modes is selected for the target domain, and others remain as the source domain (As shown in Figure 2(a)). We pre-train a small diffusion model on the source domain and subsequently fine-tune it on the target domain, observing distinct behaviors between CFG and DoG. Figure 2(b) reveals that CFG biases sampling paths away from high-density

centers, leading to outlier generations and loss of domain consistency. Conversely, as shown in Figure 2(c), DoG leverages dense pre-trained data to guide samples accurately toward high-density areas of the target domain, thereby enhancing generation quality. Details of the setup are provided in Appendix C.

**Theoretical insights into DoG.** Beyond empirical observation, we present theoretical insights into DoG, conceptualizing it as an augmentation of classifier guidance (Dhariwal & Nichol, 2021) to the conventional CFG sampling steps:

**Proposition 1.**

$$\nabla_{\mathbf{x}_t} \log p_w^{\text{DoG}}(\mathbf{x}_t|c, \mathcal{D}^{\text{tgt}}) = \nabla_{\mathbf{x}_t} \log p_w^{\text{CFG}}(\mathbf{x}_t|c, \mathcal{D}^{\text{tgt}}) + (w-1)\nabla_{\mathbf{x}_t} \log p(\mathcal{D}^{\text{tgt}}|\mathbf{x}_t) \quad (6)$$

The details can be found in Appendix B. This adjustment means that the DoG sampling distribution $p_w^{\text{DoG}}$ is tuned to discourage sampling from out-of-distribution areas, effectively using the pre-trained domain knowledge to regularize the process and improve domain consistency:

$$\frac{p_w^{\text{DoG}}}{p_w^{\text{CFG}}} \propto p(\mathcal{D}^{\text{tgt}}|\mathbf{x}_t)^{w-1} \ll 1 \quad \text{For } \mathbf{x}_t \notin \mathcal{D}^{\text{tgt}}, \quad (7)$$

highlighting how DoG steers the sampling process toward the core of the target domain manifold, thereby avoiding low-probability regions and reducing outlier generations. Figure 2(d) visually illustrates the stark guiding distinctions between CFG and DoG, underscoring the effectiveness of DoG.

**The poor fitness of the guiding model.** The second reason that limits the performance of CFG is the bad fitness of the guiding model. The low-data regime of the target domain, the conflicts arising from unconditional objectives, along with only a small slice of the training budget, result in poor fitness of the fine-tuned model Chen et al. (2023); Zhang et al. (2024). The visual quality difference is obvious if we simply inspect the unconditional images generated by the fine-tuned model. Furthermore, the unconditional case tends to work so poorly that the corresponding quantitative numbers are hardly ever reported. For example, the fine-tuned DiT-XL/2 with Stanford Car exhibits an FID of 6.57 in conditional settings versus 22.8 unconditionally.

**Theorem 1.** *Denote the mariginal distribution at timestep $t$ conditioning on $N$ data samples $\mathcal{D} = \{\boldsymbol{y}_i\}_{i=1}^N$ as $\hat{p}_t(\mathbf{x}_t) = \sum_{i=1}^N \frac{1}{N}q(\mathbf{x}_t|\mathbf{x}_0 = \boldsymbol{y}_i)$, with $\boldsymbol{y}_i \sim p(\boldsymbol{y})$. Denote the true marginal distribution at $t$ as $p_t^*(\mathbf{x}_t) = \int_{\boldsymbol{y}} p(\boldsymbol{y})q(\mathbf{x}_t|\mathbf{x}_0 = \boldsymbol{y})$. The forward process is defined as $q(\mathbf{x}_t|\mathbf{x}_0 = \boldsymbol{y}) = \mathcal{N}\left(\mathbf{x}_t|\sqrt{\bar{\alpha}_t}\boldsymbol{y}; \bar{\beta}_t\mathbf{I}\right)$. Consider the expected estimation error between $\hat{p}_t$ and $p_t^*$ with respect to all the datasets $\mathcal{D} \sim p(\mathcal{D})$, we have for all $\mathbf{x}_t$:*

$$\mathbb{E}_{\mathcal{D}\sim p(\mathcal{D})}\left[|\hat{p}_t(\mathbf{x}_t) - p_t^*(\mathbf{x}_t)|\right] \leq \frac{1}{\sqrt{N}}.$$

*Proof.* See Appendix B

**Ramark.** Given that $N^{\text{tgt}} \ll N^{\text{src}}$, it can be assumed that across most of the manifold, $\epsilon_{\boldsymbol{\theta}_0}(x)$ offers a better approximation to the ground truth marginal distribution, particularly in areas outside the target domain. In scenarios where a pre-trained model is transferred to a domain lacking in training samples, fine-tuning enhances performance on in-distribution targets but often falters on out-of-distribution samples Kumar et al. (2022). This propensity leads to a sequence of errors in diffusion model sampling, further compounding the issues faced during the guidance process. By incorporating DoG, we mitigate these errors, steering the model away from out-of-domain areas and substantially improving sampling accuracies.

## 4 EXPERIMENTS

We evaluate DoG on seven well-established fine-grained downstream datasets, comparing generation quality against standard fine-tuning with CFG. Additionally, we conduct comprehensive experiments to analyze the specific properties of each component within DoG. Detailed implementation information can be found in Appendix A.

Table 1: Comparisons on downstream tasks with pre-trained DiT-XL-2-256x256. FID ↓

| Dataset / Method | Food | SUN | Caltech | CUB Bird | Stanford Car | DF-20M | ArtBench | Average FID |
|---|---|---|---|---|---|---|---|---|
| Fine-tuning (w/o guidance) | 16.04 | 21.41 | 31.34 | 9.81 | 11.29 | 17.92 | 22.76 | 18.65 |
| + Classifier-free guidance | 10.93 | 14.13 | 23.84 | 5.37 | 6.32 | 15.29 | 19.94 | 13.69 |
| **+ Domain guidance** | **9.25** | **11.69** | **23.05** | **3.52** | **4.38** | **12.22** | **16.76** | **11.55** |
| Relative promotion | 15.36% | 17.27% | 3.31% | 34.45% | 30.70% | 20.08% | 15.95% | 19.59% |

Table 2: Comparisons on downstream tasks with pre-trained DiT-XL-2-256x256. $FD_{DINOv2}$ ↓

| Dataset / Method | Food | SUN | Caltech | CUB Bird | Stanford Car | DF-20M | ArtBench | Average $FD_{DINOv2}$ |
|---|---|---|---|---|---|---|---|---|
| Fine-tuning (w/o guidance) | 626.90 | 796.77 | 551.69 | 421.29 | 351.97 | 594.50 | 337.87 | 501.48 |
| + Classifier-free guidance | 423.90 | 653.19 | 416.78 | 198.12 | 219.25 | 326.77 | 291.23 | 363.58 |
| **+ Domain guidance** | **351.93** | **620.58** | **392.92** | **140.00** | **134.15** | **151.39** | **257.39** | **292.62** |
| Relative promotion | 20.0% | 5.0% | 5.7% | 29.3% | 38.8% | 53.7% | 11.62% | 23.4% |

**Setup.** Fine-tuning a pre-trained diffusion model to a target downstream domain is a fundamental task in transfer learning. We utilize the publicly available pre-trained model DiT-XL/2[1] (Peebles & Xie, 2023), which is pre-trained on ImageNet at a resolution of $256 \times 256$ for 7 million training steps, achieving a Fréchet Inception Distance (FID) of 2.27 (Heusel et al., 2017). Our benchmark setups include 7 fine-grained downstream datasets: Food101 (Bossard et al., 2014), SUN397 (Xiao et al., 2010), DF20-Mini (Picek et al., 2022), Caltech101 (Griffin et al., 2007), CUB-200-2011 (Wah et al., 2011), ArtBench-10 (Liao et al., 2022), and Stanford Cars (Krause et al., 2013). Most of these datasets are selected from CLIP downstream tasks except ArtBench-10 and DF-20M. DF-20M has no overlap with ImageNet, while ArtBench-10 features a distribution that is completely distinct from ImageNet. This diversity allows for a more comprehensive evaluation of DoG in scenarios where pre-trained data are significantly different from the target domain. We perform fine-tuning for 24,000 steps with a batch size of 32 at $256 \times 256$ resolution for all benchmarks. The standard fine-tuned models are trained in a CFG style, with a label dropout ratio of 10%. Each fine-tuning task is executed on a single NVIDIA A100 40GB GPU over approximately 6 hours. Following prior evaluation protocols (Peebles & Xie, 2023; Xie et al., 2023), we generate 10,000 images with 50 sampling steps per benchmark, setting the guidance weights for both CFG and DoG to 1.5. We calculate metrics between the generated images and a test set, reporting the widely used FID[2] (Heusel et al., 2017) and the more recent $FD_{DINOv2}$[3] (Stein et al., 2024a) for a richer evaluation. More detailed results of precision and recall can be found in Appendix D.

**Results** The FID results are summarized in Table 1, while the $FD_{DINOv2}$ results are presented in Table 2. Our results indicate that standard CFG is crucial for class-conditional generation, despite its inherent challenges. In contrast, the proposed DoG consistently improves all transfer tasks, effectively addressing the limitations faced by CFG and significantly enhancing generation quality—resulting in a relative FID improvement of 19.59% compared to CFG. Notably, the last two columns for DF20-Mini and ArtBench-10 exhibit a significant discrepancy from the pre-trained domain. Despite this challenge, DoG performs well, showcasing its robustness in various transfer scenarios. Even when the guided pre-trained model is considerably distant from the target domain, DoG effectively steers the generation process, enhancing both domain alignment and overall generation quality. This capability underscores DoG's utility in bridging substantial gaps between pre-trained and target domains, ensuring consistent performance across diverse settings.

## 4.1 ABLATION STUDY AND DISCUSSION

**Discussion on unconditional guiding models.** Building on the discussion of unconditional guiding models presented in Section 3.3, a pertinent question arises: Can extending the training budget

---

[1]https://dl.fbaipublicfiles.com/DiT/models/DiT-XL-2-256x256.pt

[2]https://github.com/mseitzer/pytorch-fid

[3]https://github.com/layer6ai-labs/dgm-eval

Table 3: Results of CFG and DoG on varying sampling steps. FID ↓

| Steps | CUB bird | | SUN | |
|---|---|---|---|---|
| | CFG | DoG | CFG | DoG |
| 25 | 9.69 | **4.60** | 24.34 | **19.87** |
| 50 | 5.37 | **3.52** | 14.13 | **11.69** |
| 100 | 4.27 | **3.35** | 10.07 | **8.71** |

Table 4: Results of DoG on varying training strategies. FID ↓

| Training steps | Dropout | ArtBench | Caltech | DF20M |
|---|---|---|---|---|
| 24,000 | ✓ | 16.76 | 23.05 | 12.22 |
| 21,600 | ✗ | 16.33 | 22.93 | 11.83 |
| 24,000 | ✗ | 16.13 | 22.44 | 11.60 |

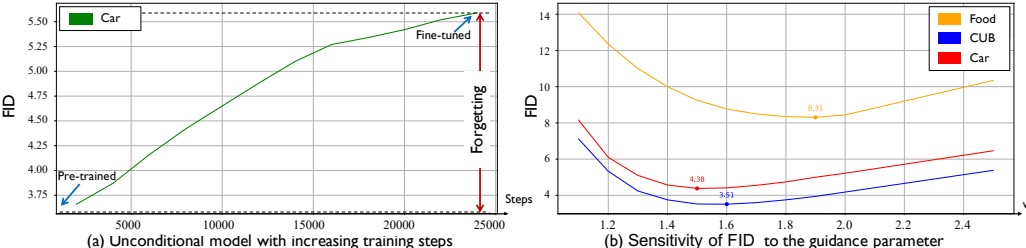

Figure 3: Component analysis of DoG. (a) illustrates that a separately fine-tuned unconditional guiding model degrades generation performance as training steps increase. (b) shows the sensitivity of FID to guidance parameters in DoG.

for a separate unconditional model address this issue? To explore this, we conducted an analysis using the CUB dataset. We fine-tuned separate unconditional guiding models with varying numbers of fine-tuning steps and employed them to guide the fine-tuned conditional model. As illustrated in Figure 3(a), this approach is ineffective—performance actually deteriorates as the number of fine-tuning steps increases. This counterintuitive outcome can be attributed to catastrophic forgetting and overfitting, where the model loses valuable pre-trained knowledge and becomes overly focused on the target domain in a low-data regime, thereby diminishing the effectiveness of the guidance.

**Discarding the unconditional training.** As previously noted, DoG focuses exclusively on modeling the class conditional density without the necessity of jointly fitting an unconditional guiding model. To illustrate this, we present a comparison of different training strategies in Table 3 using the Caltech101, DF20M, and ArtBench datasets. The dropout ratio $\delta$ is set to $10\%$ when applicable; otherwise, it is set to $0$, indicating no unconditional training involved. The first row displays the results from standard fine-tuning in a CFG style (with DoG results reported in Table 1). The second row corresponds to the same 21,600 conditional training steps (90% of the standard fine-tuning). The third row demonstrates that, under the same computational budget, DoG yields superior results. The improvement observed in the second row suggests that eliminating conflicts arising from the multi-task training of the unconditional guiding model is advantageous, along with achieving a 10% reduction in training costs.

**Sensitivity of the guidance weight factor.** Figure 3(b) probs the sensitivity to guidance weight factor in DoG across various datasets. Our best results are typically achieved with values of $w$ ranging from $1.4$ to $2$, indicating a relatively narrow search space for this parameter. To ensure fair comparisons with limited resources, all other results reported in this paper are fixed at $w = 1.5$.

**Sampling steps.** We also evaluate DoG using various sampling parameters, typically halving and doubling the default 50 sampling steps employed in iDDPM (Nichol & Dhariwal, 2021). Table 3 shows a consistent improvement in performance across these configurations. Notably, the guidance signal from DoG demonstrates a more significant enhancement with fewer steps, suggesting that the guidance becomes more precise due to the reduced variance provided by DoG.

## 4.2 QUALITATIVE RESULTS

Figure 4 showcases examples of generated images for fine-tuned downstream tasks as listed in Table 1. These examples demonstrate that both CFG and DoG enhance the perceptual quality of images, with clearer outcomes as the guidance weight increases. However, CFG, hampered by its insufficient

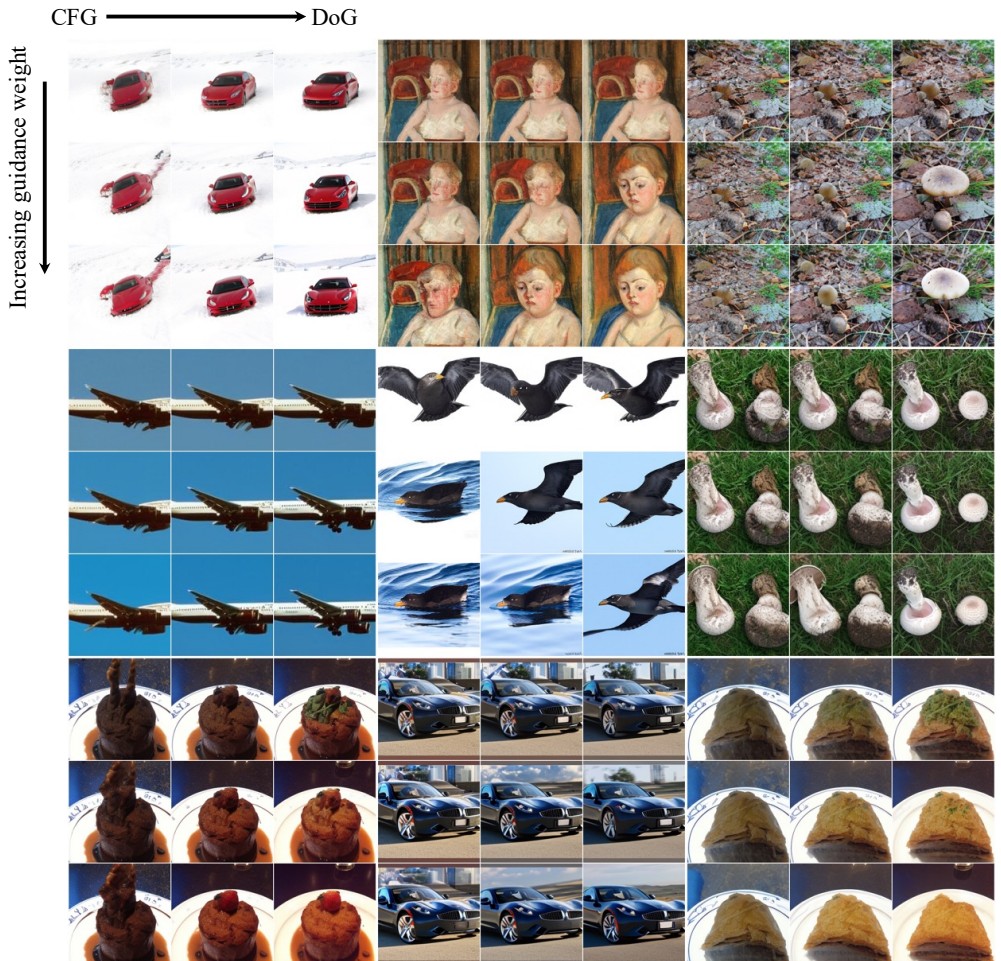

Figure 4: Qualitative showcases for DoG across downstream tasks. *Best viewed zoomed in.* Each nine-grid case compares CFG (left column) and DoG (right column), with the middle column blending the two. Rows increase guidance weights from $\{2, 3, 4\}$.

utilization of pre-trained knowledge and the limitations of its poor unconditional guiding model, often directs the sampling process toward out-of-distribution (OOD) outliers. This misdirection can result in noticeable distortions or blurring in the generated images. In contrast, DoG effectively counters these challenges, steering the generative process towards more accurate and visually appealing representations. A notable example is seen in the depiction of an airplane in the middle-left panel of the figure. Under CFG, the airplane's fuselage appears fragmented, and this distortion intensifies as the guidance weight increases. DoG, on the other hand, maintains the integrity of the airplane's structure, producing a coherent and detailed image without the distortions observed with CFG.

## 4.3 Applying DoG to Stable Diffusion with LoRAs

To evaluate the adaptability of DoG across a broader range of models and in conjunction with LoRA fine-tuning, we conduct experiments using off-the-shelf LoRAs of the SDXL model available in the Huggingface community. Specifically, we employ the SDXL model[4] and select two off-the-shelf LoRA adapters: Chalkboard style[5] and Yarn art style[6]. In our implementation of DoG, we compute the text-conditional output using the LoRA adapters while disabling the LoRA during the computation of the unconditional output. We adopt a default guidance scale of 5.0 for each

---

[4]https://huggingface.co/stabilityai/stable-diffusion-xl-base-1.0
[5]https://huggingface.co/Norod78/sdxl-chalkboarddrawing-lora
[6]https://huggingface.co/Norod78/SDXL-YarnArtStyle-LoRA

| Prompt: | CFG | DoG |
|---|---|---|

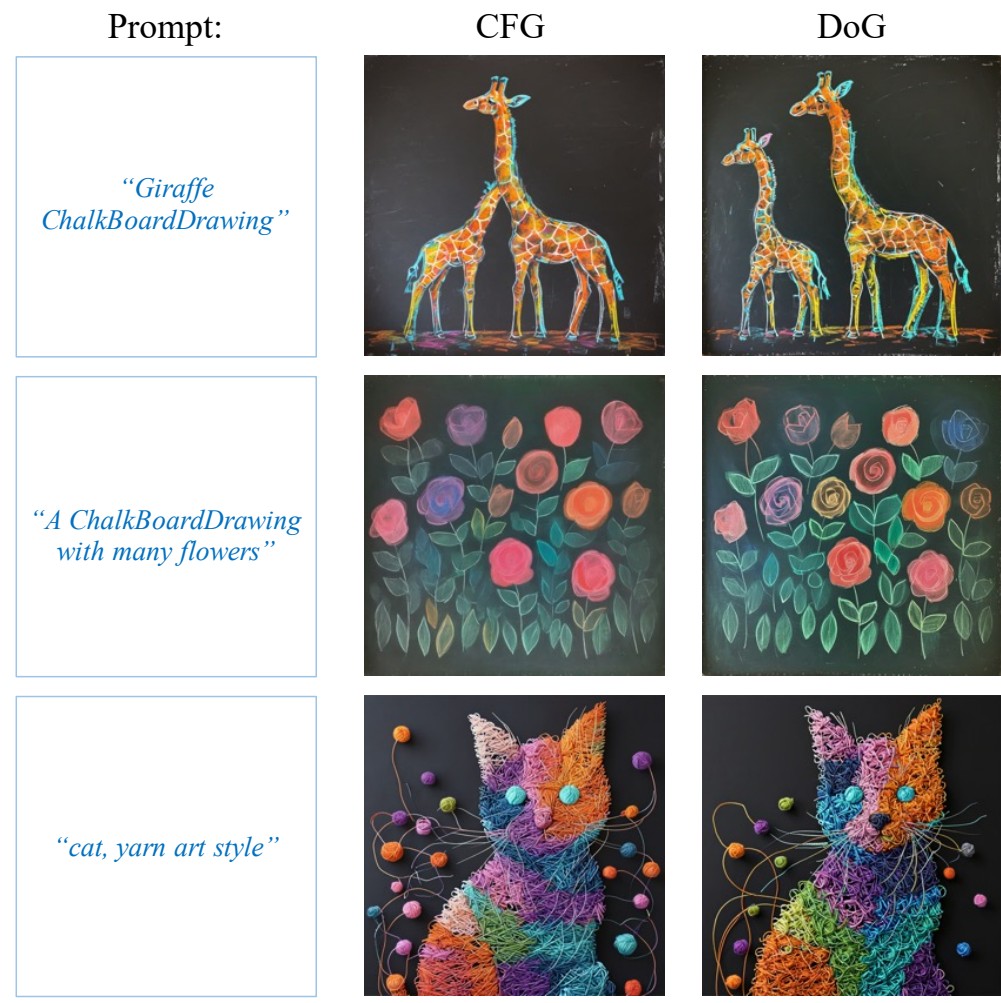

Figure 5: Qualitative showcases for LoRA-based transfer tasks with SDXL. The first two rows feature showcases in Chalkboard style transfer tasks, and the bottom row shows the Yarn art style transfer task. (A default guidance scale of 5.0 for each generation)

generation. The qualitative results, as showcased in Figure 5, indicate that DoG can produce more vivid and contextually enriched generations compared to CFG. To quantitatively assess our method, we calculate the CLIP score between the generated images and the prompts in the fine-tuning dataset. In the ChalkboardDrawing Style task, the CLIP Score increases from 27.23 with CFG to 35.24 with DoG. In the Yarn Art style, the CLIP Score increases from 34.89 to 35.03. These results demonstrate that DoG seamlessly adapts to pre-trained text-to-image models and integrates effectively with LoRA-based fine-tuning. Additional qualitative comparisons can be found in Appendix F.

## 5    CONCLUSION AND FUTURE WORKS

In this paper, we provide a novel conditional generation perspective for the transfer of pre-trained diffusion models. Based on this viewpoint, we introduce *domain guidance*, a simple transfer approach in a similar format of classifier-free guidance, improving the transfer performance significantly. We provide both empirical and theoretical evidence revealing that the effectiveness of the DoG stems from leveraging the knowledge of the pre-trained model to improve domain consistency and reduce OOD accumulated error in the sampling process. Given the promising results in this paper, potential future work could further explore the compositional guiding model for transfer learning or study a general large-scale pre-trained model serving as a unified guiding model, improving the transfer performance in arbitrary downstream tasks.

ACKNOWLEDGMENTS

This work was supported by the National Natural Science Foundation of China (U2342217 and 62021002), the BNRist Project, and the National Engineering Research Center for Big Data Software.

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

## A  IMPLEMENTATION DETAILS

In this section we provide the details of our experiments. All of our experiments are inplemented using PyTorch and conducted on NVIDIA A100 40G GPUs.

### A.1  BENCHMARK DESCRIPTION

This section describes the benchmarks used for finetuning.

**Food101 (Bossard et al., 2014)** This dataset consists of 101 food categories with a total of 101,000 images. For each class, 750 training images preserving some amount of noise and 250 manually reviewed test images are provided. All images were rescaled to have a maximum side length of 512 pixels.

**SUN397 (Xiao et al., 2010)** The SUN397 dataset contains 108,753 images of 397 well-sampled categories from the origin Scene UNderstanding (SUN) database. The number of images varies across categories, but there are at least 100 images per category. We finetune our domain model on a random partition of the whole dataset with 76,128 training images, 10,875 validation images and 21,750 test images.

**DF20M (Picek et al., 2022)** Danish Fungi 2020 (DF20) is a new fine-grained dataset and benchmark featuring highly accurate class labels based on the taxonomy of observations submitted to the Danish Fungal Atlas. The dataset has a well-defined class hierarchy and a rich observational metadata. It is characterized by a highly imbalanced long-tailed class distribution and a negligible error rate. Importantly, DF20 has no intersection with ImageNet, ensuring unbiased comparison of models fine-tuned from ImageNet checkpoints.

**Caltech101 (Griffin et al., 2007)** The Caltech 101 dataset comprises photos of objects within 101 distinct categories, with roughly 40 to 800 images allocated to each category. The majority of the categories have around 50 images. Each image is approximately 300×200 pixels in size.

**CUB-200-201 (Griffin et al., 2007)** CUB-200-2011 (Caltech-UCSD Birds-200-2011) is an expansion of the CUB- 200 dataset by approximately doubling the number of images per category and adding new annotations for part locations. The dataset consists of 11,788 images divided into 200 categories.

**ArtBench10 (Liao et al., 2022)** ArtBench-10 is a class-balanced, standardized dataset comprising 60,000 high- quality images of artwork annotated with clean and precise labels. It offers several advantages over previous artwork datasets including balanced class distribution, high-quality images, and standardized data collection and pre-processing procedures. It contains 5,000 training images and 1,000 testing images per style.

**Oxford Flowers (Nilsback & Zisserman, 2008)** The Oxford 102 Flowers Dataset contains high quality images of 102 commonly occurring flower categories in the United Kingdom. The number of images per category range between 40 and 258. This extensive dataset provides an excellent resource for various computer vision applications, especially those focused on flower recognition and classification.

**Stanford Cars (Krause et al., 2013)** In the Stanford Cars dataset, there are 16,185 images that display 196 distinct classes of cars. These images are divided into a training and a testing set: 8,144 images for training and 8,041 images for testing. The distribution of samples among classes is almost balanced. Each class represents a specific make, model, and year combination, e.g., the 2012 Tesla Model S or the 2012 BMW M3 coupe.

## A.2 EXPERIMENT DETAILS

For all our experiments, we use the ImageNet pre-trained DiT-XL/2 (Peebles & Xie, 2023) as the unconditional model to provide the guidance in DoG. For finetuning on domain, we provide the hyperparameter configuration below:

Table 5: Hyperparameter of domain transfer experiments

| Hyperparameter | Configuration |
|---|---|
| Backbone | DiT-XL/2 |
| Image Size | 256 |
| Batch Size | 32 |
| Learning Rate | 1e-4 |
| Optimizer | Adam |
| Training Steps | 24,000 |
| Validation Interval | 24,000 |
| Sampling Steps | 50 |

## B PROOFS OF THEORETICAL EXPLANATION IN SECTION 3.3

**Theorem 2** (Full version of Proposition 1). *Denote*

$$\nabla_{\mathbf{x}_t} \log p_w^{\text{DoG}}(\mathbf{x}_t|c, \mathcal{D}^{\text{tgt}}) := \nabla_{\mathbf{x}_t} \log p(\mathbf{x}_t|c, \mathcal{D}^{\text{tgt}}) + (w-1)\left(\nabla_{\mathbf{x}_t} \log p(\mathbf{x}_t|c, \mathcal{D}^{\text{tgt}}) - \nabla_{\mathbf{x}_t} \log p(\mathbf{x}_t)\right)$$

*as the underlying score function corresponding to domain guidance, and let*

$$\nabla_{\mathbf{x}_t} \log p_w^{\text{CFG}}(\mathbf{x}_t|c, \mathcal{D}^{\text{tgt}}) := \nabla_{\mathbf{x}_t} \log p(\mathbf{x}_t|c, \mathcal{D}^{\text{tgt}}) + (w-1)\left(\nabla_{\mathbf{x}_t} \log p(\mathbf{x}_t|c, \mathcal{D}^{\text{tgt}}) - \nabla_{\mathbf{x}_t} \log p(\mathbf{x}_t|\mathcal{D}^{\text{tgt}})\right)$$

*denote the score function of CFG. Then domain guidance is equivalent to applying classifier guidance to the target domain:*

$$\nabla_{\mathbf{x}_t} \log p_w^{\text{DoG}}(\mathbf{x}_t|c, \mathcal{D}^{\text{tgt}}) = \nabla_{\mathbf{x}_t} \log p_w^{\text{CFG}}(\mathbf{x}_t|c, \mathcal{D}^{\text{tgt}}) + (w-1)\nabla_{\mathbf{x}_t} \log p(\mathcal{D}^{\text{tgt}}|\mathbf{x}_t) \quad (8)$$

*Proof.* Since

$$\nabla_{\mathbf{x}_t} \log p_w^{\text{DoG}}(\mathbf{x}_t|c, \mathcal{D}^{\text{tgt}}) - \nabla_{\mathbf{x}_t} \log p_w^{\text{CFG}}(\mathbf{x}_t|c, \mathcal{D}^{\text{tgt}}) = (w-1)\left(\nabla_{\mathbf{x}_t} \log p(\mathbf{x}_t|\mathcal{D}^{\text{tgt}}) - \nabla_{\mathbf{x}_t} \log p(\mathbf{x}_t)\right)$$

Using Bayes' rule, we have:

$$\frac{p(\mathbf{x}_t|\mathcal{D}^{\text{tgt}})}{p(\mathbf{x}_t)} \propto p(\mathcal{D}^{\text{tgt}}|\mathbf{x}_t)$$

Thus we have:

$$\nabla_{\mathbf{x}_t} \log p(\mathbf{x}_t|\mathcal{D}^{\text{tgt}}) - \nabla_{\mathbf{x}_t} \log p(\mathbf{x}_t) = \nabla_{\mathbf{x}_t} \log p(\mathcal{D}^{\text{tgt}}|\mathbf{x}_t)$$

$\square$

*Proof of Theorem 1.* We denote $\hat{p}_t(\mathbf{x}_t) = \sum_{i=1}^{N} \frac{1}{N} q(\mathbf{x}_t|\mathbf{x}_0 = \boldsymbol{y}_i)$ to be the marginal distribution at time $t$ conditioning on the dataset samples $\mathcal{D} = \{\boldsymbol{y}_i\}_{i=1}^{N}$, $\boldsymbol{y}_i \sim p(\boldsymbol{y})$ and $p_t^*(\mathbf{x}_t) = \int_{\boldsymbol{y}} p(\boldsymbol{y}) q(\mathbf{x}_t|\mathbf{x}_0 = \boldsymbol{y})$ is the marginal distribution of the ground truth. Then we have:

$$\mathbb{E}_{\mathcal{D}\sim p(\mathcal{D})}\left[|p_t^*(\mathbf{x}_t) - \hat{p}_t(\mathbf{x}_t)|\right]$$

$$\leq \sqrt{\mathbb{E}_{\mathcal{D}\sim p(\mathcal{D})}\left[(p_t^*(\mathbf{x}_t) - \hat{p}_t(\mathbf{x}_t))^2\right]}$$

For a dataset $\mathcal{D} = \{\boldsymbol{y}_i\}_{i=1}^{N}$, $\boldsymbol{y}_i \sim p(\boldsymbol{y})$, we have $p(\mathcal{D}) = \prod_{i=1}^{N} p(\boldsymbol{y}_i)$. Thus we have:

$$\mathbb{E}_{\mathcal{D}\sim p(\mathcal{D})}\left[(p_t^*(\mathbf{x}_t) - \hat{p}_t(\mathbf{x}_t))^2\right]$$

$$= \int_{\{\boldsymbol{y}_i\}_{i=1}^N} \prod_{i=1}^{N} p(\boldsymbol{y}_i) \left(\sum_{i=1}^{N} \frac{1}{N} q(\mathbf{x}_t|\mathbf{x}_0 = \boldsymbol{y}_i) - p_t^*(\mathbf{x}_t)\right)^2$$

$$= \int_{\{\boldsymbol{y}_i\}_{i=1}^N} \prod_{i=1}^{N} p(\boldsymbol{y}_i) \left(\sum_{i=1}^{N} \frac{1}{N} (q(\mathbf{x}_t|\mathbf{x}_0 = \boldsymbol{y}_i) - p_t^*(\mathbf{x}_t))\right)^2$$

$$= \sum_{i=1}^{N} \int_{\boldsymbol{y}_i} p(\boldsymbol{y}_i) \frac{1}{N^2} \left(q(\mathbf{x}_t|\mathbf{x}_0 = \boldsymbol{y}_i) - p_t^*(\mathbf{x}_t)\right)^2$$

$$+ \sum_{i\neq j} \int_{\boldsymbol{y}_i,\boldsymbol{y}_j} p(\boldsymbol{y}_i)p(\boldsymbol{y}_j) \frac{1}{N^2} \left(q(\mathbf{x}_t|\mathbf{x}_0 = \boldsymbol{y}_i) - p_t^*(\mathbf{x}_t)\right) \left(q(\mathbf{x}_t|\mathbf{x}_0 = \boldsymbol{y}_j) - p_t^*(\mathbf{x}_t)\right)$$

$$= \frac{1}{N^2} \sum_{i=1}^{N} \int_{\boldsymbol{y}_i} p(\boldsymbol{y}_i) \left(q(\mathbf{x}_t|\mathbf{x}_0 = \boldsymbol{y}_i) - p_t^*(\mathbf{x}_t)\right)^2 \tag{9}$$

$$\leq \frac{1}{N}, \tag{10}$$

where Eq 9 is because $\int_{\boldsymbol{y}_i} p(\boldsymbol{y}_i) \left(q(\mathbf{x}_t|\mathbf{x}_0 = \boldsymbol{y}_i) - p_t^*(\mathbf{x}_t)\right) = 0$, and Eq 10 is because $\int_{\boldsymbol{y}_i} p(\boldsymbol{y}_i) \left(q(\mathbf{x}_t|\mathbf{x}_0 = \boldsymbol{y}_i) - p_t^*(\mathbf{x}_t)\right)^2 \leq 1$. As a result:

$$\mathbb{E}_{\mathcal{D}\sim p(\mathcal{D})}\left[|p_t^*(\mathbf{x}_t) - \hat{p}_t(\mathbf{x}_t)|\right] \leq \frac{1}{\sqrt{N}}.$$

$\square$

## C  DETAILS OF THE 2D TOY EXAMPLE

We randomly generated 100 Gaussians $\mathcal{M}_s = \{\phi_i, \mu_i, \Sigma_i\}$ as the pre-train data distribution and generated 5 Gaussians $\mathcal{M}_t$ in a selected area as the distribution of the target domain. We divide the Gaussians into two classes $\mathcal{M}_{c1}$ and $\mathcal{M}_{c2}$ each occupying a selected area as the two class conditions. Given above, we can write the data density as:

$$\text{Source density } p_s(x) = \sum_{i\in\mathcal{M}_s} \phi_i \mathcal{N}(x|\mu_i, \Sigma_i),$$

$$\text{target density } p_t(x) = \sum_{i\in\mathcal{M}_t} \phi_i \mathcal{N}(x|\mu_i, \Sigma_i),$$

$$\text{class-conditional target density } p_t(x|c) = \sum_{i\in\mathcal{M}_c} \phi_i \mathcal{N}(x|\mu_i, \Sigma_i).$$

where the multivariate Gaussian distribution is defined as:

$$\mathcal{N}(x|\mu_i, \Sigma_i) = \frac{1}{\sqrt{(2\pi)^2 \det(\Sigma)}} \exp\left(-\frac{1}{2}(x - \mu)^\top \Sigma^{-1}(x - \mu)\right)$$

We implement the denoising network as a 4-layer fully connected ReLU network with hidden feature dimension 64. We use sinusoidal positional embeddings for time conditioning as in (Ho et al., 2020), and we add the time embedding to every intermediate layer. Unlike traditional CFG where we use a dropout ratio to learn the non-conditinal distribution, here we train a separate unconditional model on $p_t(x)$ and $p_s(x)$ for guidance, similar to (Karras et al., 2024b). We parameterize the network to explicitly output the score function. The pre-train model converges after 10000 Adam steps with a batch size of 128 and learning rate 1e-3. For fine-tuning on the target domain, the model converges after 1000 steps.

For training, we used the DDPM noise schedule from Ho et al. (2020). We used the DDIM sampler Song et al. (2020a) for 20 sampling steps to generate our samples. For all of our experiments, we set the CFG weight and the DoG weight at 2.

## D   ADDITIONAL EXPERIMENT RESULTS

Here we provide the additional results for the Precision and Recall metrics (Kynkäänniemi et al., 2019). Notably, DoG enhances precision without compromising recall, indicating an overall improvement in generation quality.

Table 6: Precision ↑ Comparisons on downstream tasks with pre-trained DiT-XL-2-256x256.

| Dataset / Method | Food | SUN | Caltech | CUB Bird | Stanford Car | DF-20M | ArtBench | Average Precision |
|---|---|---|---|---|---|---|---|---|
| Fine-tuning (w/o guidance) | 0.376 | 0.583 | 0.536 | 0.143 | 0.331 | 0.502 | 0.821 | 0.470 |
| + Classifier-free guidance | 0.455 | 0.590 | 0.668 | 0.331 | 0.501 | 0.537 | 0.831 | 0.559 |
| **+ Domain guidance** | **0.533** | **0.601** | **0.715** | **0.431** | **0.631** | **0.708** | **0.901** | **0.646** |

Table 7: Recall ↑ Comparisons on downstream tasks with pre-trained DiT-XL-2-256x256.

| Dataset / Method | Food | SUN | Caltech | CUB Bird | Stanford Car | DF-20M | ArtBench | Average Recall |
|---|---|---|---|---|---|---|---|---|
| Fine-tuning (w/o guidance) | **0.652** | 0.326 | **0.650** | **0.960** | 0.840 | **0.712** | 0.212 | **0.621** |
| + Classifier-free guidance | 0.640 | **0.370** | 0.548 | 0.890 | 0.840 | 0.711 | **0.230** | 0.604 |
| **+ Domain guidance** | 0.651 | **0.370** | 0.546 | 0.860 | 0.840 | 0.638 | **0.230** | 0.590 |

We conducted experiments applying DoG to off-the-shelf LoRAs of the SDXL model available in the Huggingface community. The results in Table 8 show that DoG can enhance the CLIP Score of the fine-tuned model significantly.

Table 8: CLIP Score ↑ Enhance the transfer of the SDXL model with the off-the-shelf LoRAs

| CLIP Score ↑ | Chalkboard Drawing Style ↑ | Yarn Art Style ↑ |
|---|---|---|
| Real data | 36.02 | 33.88 |
| Off-the-shelf LoRAs with CFG | 27.23 | 34.89 |
| Off-the-shelf LoRAs with DoG | **35.24** | **35.03** |

Table 9: FID ↓ Transferring pre-trained DiT-XL-2-512x512 to Food 512x512 dataset

| Food (512x512) | FID ↓ | Relative Promotion ↑ |
|---|---|---|
| Fine-tuning with CFG | 13.56 | 0.0% |
| Fine-tuning with DoG | **11.05** | **18.5%** |

Table 10: FID ↓ Comparision with different pairs of pre-trained and guiding models

| CUB 256x256 FID ↓ | Fine-tune DiT-L/2-300M ↓ | Fine-tune DiT-XL/2-700M ↓ |
|---|---|---|
| Standard CFG | 15.20 | 5.32 |
| DoG with DiT-L-2-300M | **6.56** | 3.85 |
| DoG with DiT-XL-2-700M | 8.80 | **3.52** |

Table 11: FID ↓ Incorporate with DiffFit

| Transfer DiT-XL/2 to CUB 256x256 | FID ↓ |
|---|---|
| DiffFit with CFG | 4.98 |
| DiffFit with DoG | **3.66** |

## E    RELATIONS WITH AUTOGUIDANCE

Autoguidance (Karras et al., 2024a) focuses on guiding models from suboptimal outputs to better generations, typically using a less-trained version of the model for guidance, equipped with guidance interval techniques (Kynkäänniemi et al., 2024). This method has demonstrated remarkable performance in EDM group methods.

DoG and Autoguidance originate from different contexts: Autoguidance focuses on improving generation within the same domain, whereas DoG is designed to adapt pre-trained models to domains outside their original training context. It is inappropriate to consider the original pre-trained model in DoG as a suboptimal version of the fine-tuned model.

Furthermore, pre-trained models are usually optimized on extensive datasets with distinct training strategies, whereas fine-tuned models are trained on different downstream domains with limited data points. This difference undermines the same degradation hypothesis posited by Autoguidance.

Additionally, the pre-trained model is generally a well-trained model with rich knowledge, while fine-tuning to a small dataset often leads to catastrophic forgetting and poor fitness of the target domain. It is not accurate to state that the pre-trained model is a bad version of the fine-tuned one.

As demonstrated in Table 10, using a better DiT-XL/2 to guide the model fine-tuned from DiT-L/2 conceptually achieves the transfer gain.

Autoguidance is mainly proved in EDM context, whereas DoG mainly provides evidence in DiT-based model and text2img stable diffusion models. Exploring a general understanding of guidance is indeed an valuable avenue for future work. We believe it is valuable to formally establish the properties required for a unified unconditional guide model and to develop a general guidance model beneficial to all diffusion tasks.

# F    ADDITIONAL QUALITATIVE SAMPLES WITH OFF-THE-SHELF LORAS

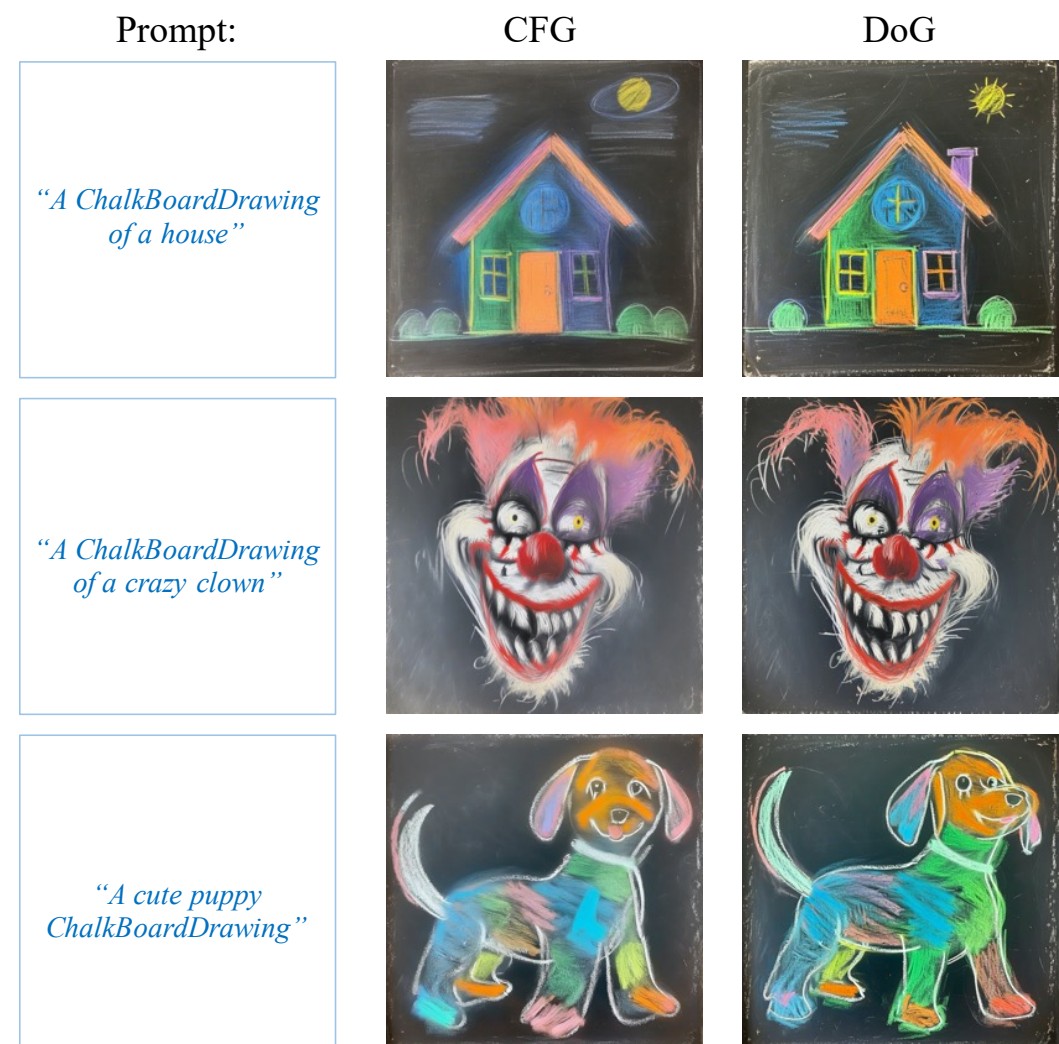

Figure 6: Qualitative showcases of the Chalkboard style transfer task, utilizing a default guidance scale of 5.0 for each generation.

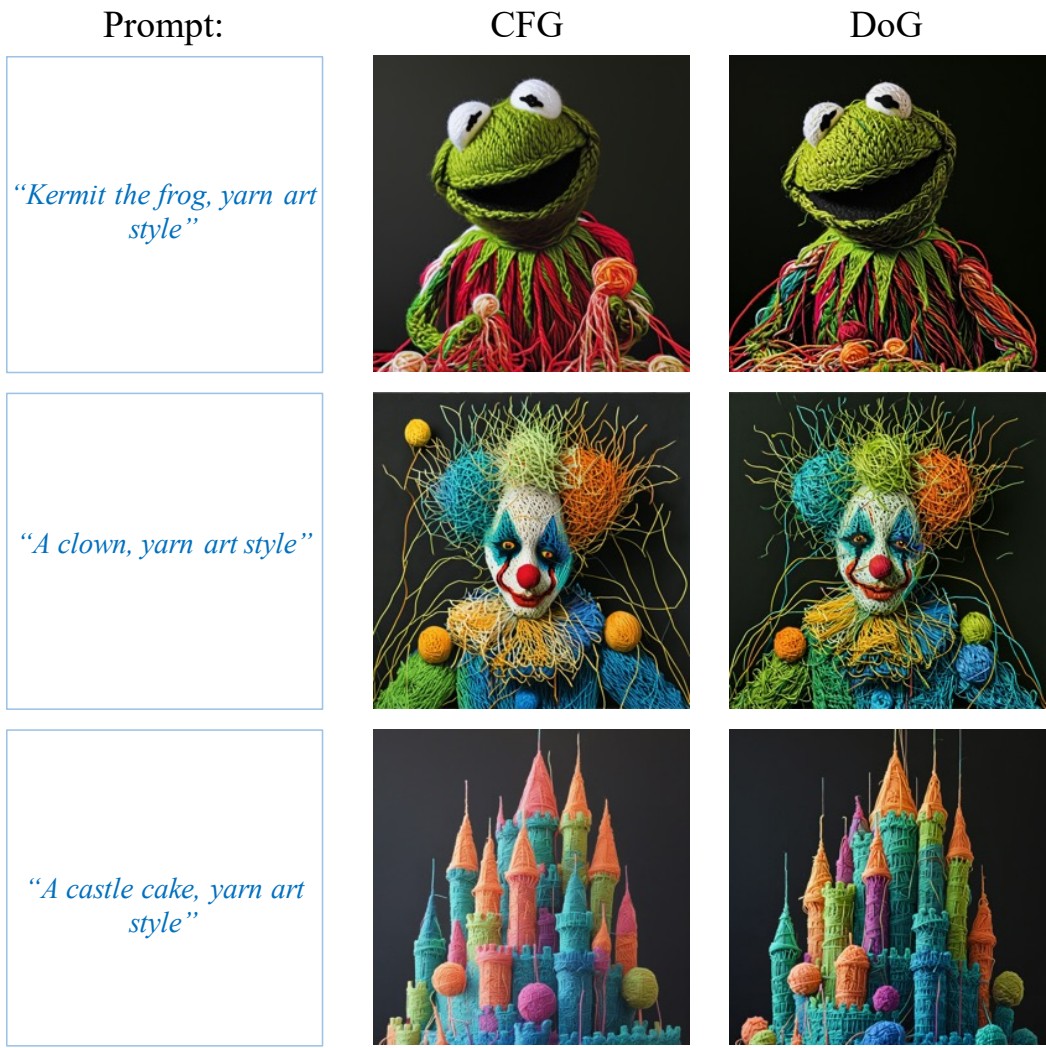

Figure 7: Qualitative showcases of the Yarn art style transfer task, utilizing a default guidance scale of 5.0 for each generation.

