# OpenReview forum: "Domain Guidance: A Simple Transfer Approach for a Pre-trained Diffusion Model"
_ICLR.cc/2025/Conference — ICLR 2025 Poster_

### Official Review · Reviewer_Pu4u · 2024-10-31

**Soundness:** 3
**Presentation:** 3
**Contribution:** 3
**Rating:** 6
**Confidence:** 3

**Summary:**

This work leverages pre-trained knowledge as the domain guidance to guide the model toward the target domain. This work thinks that domain guidance facilitates better domain alignment and higher-quality generations, and provides correlation theoretical analyses.

**Strengths:**

Originality: The idea of using the pre-trained model as a domain guidance to guide transfer learning is innovative.

Quality and Clarity: The quality of this is satisfactory and the analysis is thoughtful. The paper is well-written, with a thorough and comprehensive ablation study.

Significance. This work provides a contribution to the transfer learning of diffusion learning. In particular, this work can be seamlessly integrated into the existing fine-tuning models without additional training.

**Weaknesses:**

1. Is there a domain gap between personalized scenarios and general scenarios?
2. During the transfer learning process, does all the knowledge from the pre-trained model need to be effectively utilized?
3. How can the impact of the knowledge in the pre-trained model be assessed in the target domain?
4. For DF-20M (which has no overlap with ImageNet) and ArtBench-10 (whose feature distribution is completely distinct from ImageNet), why can guidance from the model pre-trained on ImageNet lead to better generation?
5. Is the distribution of the downstream domain a subset of the distribution of the pre-trained domain?

**Questions:**

it is the same as Weaknesses.

---

> ### Author Response · Authors · 2024-11-24
>
> We sincerely thank the reviewer for the efforts in reviewing our paper. Our responses according to the reviewer's questions are summarized as follows.
>
> > W1:  Is there a domain gap between personalized scenarios and general scenarios
>
> Yes, absolutely. From a distributional perspective, the data distributions of the personalized domain (downstream target domain) and the general domain (pre-trained source domain) differ in two key aspects: the means of the distributions are distinct, and the variance in the personalized domain is typically smaller than that of the general domain. In our experiments, the domain gap can be clearly identified in the selected transfer tasks. For example, the CUB dataset includes fine-grained birds, and ArtBench-10 consists of images of artwork, while the pre-trained source domain, ImageNet, is collected from search engines and annotated into 1000 categories.
>
> >W2: During the transfer learning process, does all the knowledge from the pre-trained model need to be effectively utilized?
>
> No, not all knowledge from the pre-trained model is necessary for individual domains or tasks. Some knowledge may be domain-specific, while other knowledge can be beneficial across different domains, thereby improving performance for the target domain or task. In cases where the source and target domains are unrelated, brute-force transfer may be unsuccessful. Transfer learning focuses on transferring the beneficial parts of the knowledge from the pre-trained model across domains. Domain Guidance (DoG) provides novel techniques that leverage this pre-trained knowledge through unconditional guidance.
>
> > W3: How can the impact of the knowledge in the pre-trained model be assessed in the target domain?
>
> The impact of the pre-trained model in DoG consists of two parts: the first is the faster convergence and improved generation quality derived from the full fine-tuning process compared to training from scratch. The second part is brought by DoG itself, which can be assessed by the significant improvement over the standard Classifier-Free Guidance method. Generally, pre-trained models are well-trained on much larger datasets and tend to exhibit good generalization capabilities. However, these capabilities are often diminished in the fine-tuning process due to the catastrophic forgetting phenomenon. DoG utilizes this knowledge by directly incorporating outputs from the pre-trained model in the sampling process to enhance generation quality in the target domain.
>
> > W4: For DF-20M (which has no overlap with ImageNet) and ArtBench-10 (whose feature distribution is completely distinct from ImageNet), why can guidance from the model pre-trained on ImageNet lead to better generation?
>
> Although these domains are quite different in appearance, some beneficial common knowledge can still be leveraged. In the guidance process, the pre-trained models serve as the unconditional branch,  from which the generation process is pushed away. When the domain gap is large, the fine-tuned model may struggle to fit the unconditional branch of the target domain due to limited optimization steps, during which we reduce the conditional signal. DoG provides a mechanism to leverage pre-trained knowledge to assist the generation process far from the pre-trained domain, a point supported by the improvements in our experimental results.
>
> >W5: Is the distribution of the downstream domain a subset of the distribution of the pre-trained domain?
>
> No, it is not nessissary that the domainstream domain is a subset of the distribution of the pre-trained domain.
>
> ----
>
> We hope that our additional clarifications and discussion address your questions and concerns. Please let us know if we miss some of your point or you have any further concerns!

---

> > ### Comment · Reviewer_Pu4u · 2024-12-02
> > **I'll update the score.**
> >
> > I appreciated the authors' response. I think that my concerns and the other reviews' concerns were answered well, so I updated the score

---

> ### Author Response · Authors · 2024-11-27
> **Eagerly Await Your Response**
>
> Dear Reviewer Pu4u,
>
> Thank you once again for your valuable and constructive feedback. As the author-reviewer discussion period is drawing to a close, we kindly request your feedback on our rebuttal provided above. Please let us know if our responses have addressed your concerns adequately. If there are any remaining issues, we are eager to engage in further discussion, as we believe that a thorough discussion will contribute to strengthening our paper.
>
> Best regards,
>
> Authors

---

### Official Review · Reviewer_CXsS · 2024-11-04

**Soundness:** 3
**Presentation:** 4
**Contribution:** 4
**Rating:** 8
**Confidence:** 4

**Summary:**

This submission proposed a new conditioning method to transfer a diffusion model
 to a new domain named Domain Guidance (DoG).
The idea behind DoG is to pull the generation process toward the target domain
from the pre-trained domain.
Given a pretrained model and fine-tuned version of it, domain guidance guides the process
to the direction that the fine-tuned one indicates and against the pretrained ones's direction.
Theoretical analyses show that DoG mitigates sampling from out-of-distribution area of the target domain.
Experiments show that DoG outperforms CFG in combination with diffusion models finetuned in relatively small
datasets.

**Strengths:**

- The motivation and idea of DoG is clearly presented, and the paper is overall easy to follow.
- Theoretical analyses and toy examples well depict the advantage of DoG.
- DoG outperformed CFG, a de facto standard method for diffusion guidance with a margin.

**Weaknesses:**

- Range of applicable models is not discussed well:
The experiments are conducted with ImageNet-pretrained  DiT-XL/2 and
but is it mandatory to use the same architecture and initialization weights to perform DoG?
Furthermore, transfering diffusion models to small datasets are often done with LoRA [a] rather than full finetuning.
Is DoG applicable to LoRA-based transfer?
[a] LoRA: Low-Rank Adaptation of Large Language Models


- Relationship with Autoguidance:
A similar idea is presented in a preprint [b] which uses less-trained version of the model for guidance,
instead of unfinetuned version in DoG.
Discussing this may be useful, while it it not mandatory due to its unpublished state and missing it should not be penalized.
[b] Guiding a Diffusion Model with a Bad Version of Itself, NeurIPS 2024 to appear.

**Questions:**

Please see Weaknesses 1.

---

> ### Author Response · Authors · 2024-11-24
> **Official Comment by Authors [Part 1/2]**
>
> We sincerely thank the reviewer for the efforts in reviewing our paper. Our responses according to the reviewer's comments are summarized as follows.
>
> > W1-1: Range of applicable models: is it manatory to use the same architecture and initialization weights?
>
> Thank you for your insightful question. We pre-trained a smaller DiT-L/2 model with 300 million iterations, while the open-source DiT-XL/2 was pre-trained with 700 million iterations on ImageNet. These models differ in size and training extent. We fine-tuned both models on the CUB (256x256) dataset to evaluate the performance of Domain Guidance (DoG) with different pairs of pre-trained and guiding models.
>
> **Table C: Comparision with different pairs of pre-trained and guiding models**
>
> | CUB 256x256 FID $\downarrow$ | Fine-tune DiT-L/2-300M $\downarrow$ | Fine-tune DiT-XL/2-700M $\downarrow$ |
> | :--------------------------- | ----------------------------------- | ----------------------------------------- |
> | Standard CFG                 | 15.20                               | 5.32                                      |
> | DoG with DiT-L-2-300M        | **6.56**                            | 3.85                                      |
> | DoG with DiT-XL-2-700M       | 8.80                                | **3.52**                                  |
>
> The results demonstrate that different pairs of pre-trained and guiding models can consistently enhance generation quality via DoG. Therefore, it is not necessary to use the same network architecture or initialization weights in DoG. However, using the same initialization appears to lead to better improvements. Despite the significant enhancements from both unconditional choices, the influence of these mismatches is relatively minor. The optimal selection of the guiding model remains an area for further exploration.
>
> Due to time constraints and the lack of suitable pre-trained LDM alternatives with the same VAE encoder or diffusion schedule, our analysis of the choice of the guidance model was limited.
>
> > W1-2: Is DoG applicable to LoRA-based transfer?
>
> Yes, DoG is applicable to LoRA-based transfer (also known as parameter-efficient fine-tuning, PEFT). Notably, LoRA-based methods, which generally preserve the original weights of the pre-trained model, can be conveniently implemented with DoG by directly disconnecting the adapter modules.
>
>
>
> **Table B: Enhancing the transferring of the SDXL model with the off-the-shelf LoRAs**
>
> | CLIP Score $\uparrow$        | Chalkboard Drawing Style [a] $\uparrow$ | Yarn Art Style [b] $\uparrow$ |
> | :--------------------------- | ----------------------------------- | ------------------------- |
> | Real data                    | 36.02                               | 33.88                     |
> | Off-the-shelf LoRAs with CFG | 27.23                               | 34.89                     |
> | Off-the-shelf LoRAs with DoG | **35.24**                           | **35.03**                 |
>
> We conducted experiments applying DoG to off-the-shelf LoRAs of the SDXL model available in the Huggingface community. The results show that DoG can enhance the CLIP Score of fine-tuned models and produce more vivid generations, as detailed in the revised paper **(Figure 5 on page 10)**.
>
> **Table D: Incorperate with DiffFit [1]**
>
> | Transfer DiT-XL/2 to CUB 256x256 | FID $\downarrow$ |
> | :------------------------------- | ---------------- |
> | DiffFit with CFG                 | 4.98             |
> | DiffFit with DoG                 | **3.66**         |
>
> Further experiments with DiffFit [1], a state-of-the-art PEFT method in the diffusion context, demonstrate that PEFT methods can also benefit from DoG.

---

> ### Author Response · Authors · 2024-11-24
> **Official Comment by Authors [Part 2/2]**
>
> > W2: Relationship with Autoguidance
>
> Thank you for your valuable suggestion. Discussing the relationship with Autoguidance [2] is indeed meaningful and provides further context for DoG. Autoguidance focuses on guiding models from suboptimal outputs to better generations, typically using a less-trained version of the model for guidance, equipped with guidance interval techniques [3]. This method has demonstrated remarkable performance in EDM group methods.
>
> DoG and Autoguidance originate from different contexts: Autoguidance focuses on improving generation within the same domain, whereas DoG is designed to adapt pre-trained models to domains outside their original training context. It is inappropriate to consider the original pre-trained model in DoG as a suboptimal version of the fine-tuned model.
>
> Furthermore, pre-trained models are usually optimized on extensive datasets with distinct training strategies, whereas fine-tuned models are trained on different downstream domains with limited data points. This difference undermines the same degradation hypothesis posited by Autoguidance.
>
> Additionally, the pre-trained model is generally a well-trained model with rich knowledge, while fine-tuning to a small dataset often leads to catastrophic forgetting and poor fitness of the target domain. It is not accurate to state that the pre-trained model is a bad version of the fine-tuned one.
>
> As demonstrated in Table C, using a better DiT-XL/2 to guide the model fine-tuned from DiT-L/2 conceptually achieves the transfer gain.
>
> Autoguidance is mainly proved in EDM context, whereas DoG mainly provides evidence in DiT-based model and text2img stable diffusion models. Exploring a general understanding of guidance is indeed an valuable avenue for future work.
> We believe it is valuable to formally establish the properties required for a unified unconditional guide model and to develop a general guidance model beneficial to all diffusion tasks.
>
>
> ----
> [a] Chalkboard style: https://huggingface.co/Norod78/sdxl-chalkboarddrawing-lora
>
> [b] Yarn art style: https://huggingface.co/Norod78/SDXL-YarnArtStyle-LoRA
>
> [1] DiffFit: Unlocking Transferability of Large Diffusion Models via Simple Parameter-Efficient Fine-Tuning
>
> [2] Guiding a Diffusion Model with a Bad Version of Itself, NeurIPS 2024 to appear.
>
> [3] Applying guidance in a limited interval improves sample and distribution quality in diffusion models, Neurips 2024 to appear.
>
> ----
>
> We hope that our clarifications address your questions and concerns. Please let us know if there are any points we have missed or if you have further inquiries!

---

> ### Author Response · Authors · 2024-11-27
> **Request of Reviewer’s attention and feedback**
>
> Dear Reviewer CXsS,
>
> Thank you again for your detailed comments, which have been invaluable in helping us improve the quality of our paper. We have made every effort to address your concerns.  Please let us know if our responses have addressed your concerns adequately. If there are any remaining issues, we are ready to engage in further discussion.
>
> Our key updates are as follows:
>
> - **New evaluation on style transfer tasks using SDXL with LoRAs**: We conduct additional experiments on SDXL style transfer tasks using off-the-shelf LoRA weights from the Huggingface community. Both quantitative and qualitative results demonstrate the flexibility and adaptability of DoG.
> - **Integration with PEFT method**:  We  further integrate DoG with a recent PEFT method, DiffFit. The results demonstrate that PEFT methods can also benefit from DoG.
> - **Discussion on applicability for different guiding model**: Aligning with the main results conducted with LDM-DiT-XL/2 settings, we newly pre-trained a DiT-L/2 for 300M iterations using its official repository, retaining the same VAE encoder. Our experiments show that DoG performs well with different architectures (DiT-L vs. DiT-XL) and does not rely on identical initialization.
> - **Relation with Autoguidance:** We cite Autoguidance and Guidance Interval, and expanded our discussion on DoG's relation to Autoguidance.
>
> These updates have been detailed in the separate response and in the revised version of our paper. We sincerely appreciate your thoughtful review and look forward to your feedback.
>
> Best regards,
>
> Authors

---

> > ### Comment · Reviewer_CXsS · 2024-11-29
> >
> > I appreciated the authors' response.
> > I think that my concerns and the other reviews' concerns were answered well, so I updated the score.

---

### Official Review · Reviewer_CbgC · 2024-11-05

**Soundness:** 2
**Presentation:** 3
**Contribution:** 3
**Rating:** 6
**Confidence:** 4

**Summary:**

The paper introduces "Domain Guidance" (DoG), a new transfer approach for pre-trained diffusion models aimed at enhancing domain alignment and generating high-quality outputs. DoG builds on classifier-free guidance (CFG) but adapts it to transfer learning by incorporating pre-trained model knowledge to guide the generative process. Through empirical and theoretical analysis, the authors demonstrate DoG’s effectiveness in maintaining pre-trained knowledge, improving image fidelity, and reducing out-of-distribution errors. The results indicate substantial improvements in several metrics (FID and FDDINOv2) across various downstream tasks compared to standard fine-tuning and CFG.

**Strengths:**

1. DoG’s design as a CFG variant is well-motivated, with clear delineation of the benefits over standard CFG. The illustrations of DoG’s guidance process effectively highlight its domain-alignment benefits.
2. Quantitative results show that DoG significantly outperforms CFG, particularly on datasets with substantial domain shifts. Qualitative results are also compelling, with DoG generating visually consistent images even with increased guidance weights.
3. The paper provides a theoretical foundation for DoG, showing how it maintains domain alignment and mitigates OOD sampling errors, supported by an illustrative analysis using Gaussian mixtures.

**Weaknesses:**

1. The paper could benefit from more discussion on DoG’s adaptability, particularly in domains vastly different from the pre-trained model’s source domain.
2. The influence of guidance weights is explored, but further sensitivity analysis could strengthen the understanding of DoG’s stability across diverse settings.
3. While diverse datasets are used, further tests on varying resolutions and different architectures could better illustrate DoG’s flexibility.

**Questions:**

Please refer to the weaknesses.

---

> ### Author Response · Authors · 2024-11-24
>
> We sincerely thank the reviewer for the careful review and the insightful suggestions. Our responses to the concerns raised are outlined below:
>
> > W1: more discussion on DoG’s adaptability, particularly in domains vastly different from the pre-trained model’s source domain.
>
> Thank you for highlighting the importance of discussing DoG's adaptability. Notably, we evaluate DoG on DF-20M [1] and Artbench10 [2] in the current version. The DF-20M dataset contains images from the Atlas of Danish Fungi, which has zero overlap with ImageNet, and Artbench10 includes images of artwork from 10 distinctive artistic styles, vastly different from the pre-trained source domain ImageNet. DoG performs surprisingly well on these two benchmarks, as shown in Tables 1 and 2.
>
> To further address your concern, we added additional experiments that transfer Stable Diffusion to stylized generation tasks to support the adaptability of DoG. These tasks are quite different from the conditional generation tasks in Table 1 and 2. The quantitative results can be found in Table B below, and the qualitative results are in the revised paper **(Figure 5 on page 10)**.
>
> > W2: The influence of guidance weights is explored, but further sensitivity analysis could strengthen the understanding of DoG’s stability across diverse settings.
>
> Thank you for your valuable suggestion. We previously provided sensitivity analysis of two hyperparameters—guidance weight and sampling steps. Here, we extend our analysis to varying resolutions (512x512) and more architectures (SDXL with LoRAs).
>
> **Table A: Transferring pre-trained DiT-XL-2-512x512 to Food 512x512 dataset.**
>
> | Food (512x512)       | FID $\downarrow$ | Relative Promotion $\uparrow$ |
> | :------------------- | :--------------- | ----------------------------- |
> | Fine-tuning with CFG | 13.56            | 0.0%                          |
> | Fine-tuning with DoG | **11.05**        | **18.5%**                     |
>
> The results show that DoG is not sensitive to resolutions. It can also improve the performance with varying resolutions.
>
> **Table B: Enhance the transfer of the SDXL model with the off-the-shelf LoRAs**
>
> | CLIP Score $\uparrow$        | Chalkboard Drawing Style $\uparrow$ | Yarn Art Style $\uparrow$ |
> | :--------------------------- | ----------------------------------- | ------------------------- |
> | Real data                    | 36.02                               | 33.88                     |
> | Off-the-shelf LoRAs with CFG | 27.23                               | 34.89                     |
> | Off-the-shelf LoRAs with DoG | **35.24**                           | **35.03**                 |
>
>
> We conducted experiments applying DoG to off-the-shelf LoRAs of the SDXL model available in the Huggingface community. The results show that DoG can enhance the CLIP Score of the fine-tuned model significantly. We also provide showcase results in the revised paper **(Figure 5 on page 10)**, which indicates that DoG produces more vivid generations.
>
>
> The expriments of stable diffusion model with LoRAs suggests that DoG do not limited to the DiT architechture and the full fine-tuning senarios.
>
> > W3: further tests on varying resolutions and different architectures
>
> As addressed in W2, we have added experiments at 512x512 resolutions and transferring stable diffusion models with LoRAs.
>
>
> [1] Danish Fungi 2020 – Not Just Another Image Recognition Dataset, WACV 2022.
>
> [2] The ArtBench Dataset: Benchmarking Generative Models with Artworks, Arxiv 2022.
>
> [3] Chalkboard style: https://huggingface.co/Norod78/sdxl-chalkboarddrawing-lora
>
> [4] Yarn art style: https://huggingface.co/Norod78/SDXL-YarnArtStyle-LoRA
>
> ----
>
> We hope that our additional clarifications and discussion address your questions and concerns. Please let us know if we miss some of your point or you have any further concerns!

---

> ### Author Response · Authors · 2024-11-27
> **Request of Reviewer’s attention and feedback**
>
> Dear Reviewer CbgC,
>
> As the author-reviewer discussion period draws to a close, we kindly seek your feedback on our rebuttal. We have earnestly addressed your concerns and would appreciate your input on whether our responses meet your expectations. If there are any remaining issues, we are eager to engage in further discussion.
>
> Thank you for your valuable time and consideration. We look forward to your response.
>
> Best regards,
>
> Authors

---

### Author Response · Authors · 2024-11-24

We would like to sincerely thank all the reviewers for providing insightful reviews and valuable comments. Your reviews are of great importance to us in improving the quality of this work. **We have revised our paper and responded to each reviewer with a separate response.**

---

### Meta-Review · Area_Chair_vs6i · 2024-12-18

**Metareview:**

This paper proposes Domain Guidance, a new method for adapting diffusion models to new domains. DoG uses pre-trained and fine-tuned models to guide generation toward the target domain, improving domain alignment and sample quality. Theoretical and experimental results show DoG outperforms traditional guidance methods, especially with small fine-tuning datasets.

The authors have implemented several key updates to strengthen the paper. For example, new evaluations on style transfer tasks were conducted using SDXL with LoRA weights from the Huggingface community, demonstrating the flexibility and adaptability of DoG through both quantitative and qualitative results. The authors also integrated DoG with the PEFT method DiffFit. These updates contribute to a more comprehensive and robust presentation of the proposed method.

**Additional Comments On Reviewer Discussion:**

Reviewers were satisfied with the authors' response, and there appear to be no remaining concerns.

---

### Decision · Program_Chairs · 2025-01-22

Accept (Poster)